# Allostatic load and its determinants in a German sample—Results from the Carla cohort

Eric Priedemann[1]*, Alexander Kluttig[1], Frank Bernhard Kraus[2], Daniel Sedding[3], Rafael Mikolajczyk[1], Amand Führer[1]

**1** Institute for Medical Epidemiology, Biometrics, and Informatics (IMEBI), Interdisciplinary Center for Health Sciences, Medical Faculty of the Martin Luther University Halle-Wittenberg, Halle, Germany, **2** Department of Laboratory Medicine, Central Laboratory, University Hospital Halle, Halle (Saale), Germany, **3** Department of Internal Medicine III, Martin Luther-University Halle-Wittenberg, Halle (Saale), Germany

* eric.priedemann@uk-halle.de

## Abstract

### Background

Allostatic load (AL) is a surrogate of the physiological response to stress and reflects the 'wear and tear' on the body. Previous studies indicated that socioeconomic and behavioral determinants influence AL, which in turn is associated with health outcomes. Therefore, AL is increasingly used to operationalize the relationship between social inequality, stress, and health outcomes. This study aimed to investigate associated factors and patterns of AL in the population over a 20-year period using data from the CARLA cohort.

### Methods

The analysis included 473 participants from the CARLA study (Cardiovascular Disease, Living and Ageing in Halle), aged 45–80 years at baseline. From recruitment in 2002 in Halle (Saale), three follow-up examinations took place until 2022. We calculated AL scores as the sum of standardized z-scores for metabolic, immune, cardiovascular, and anthropometric components. Descriptive statistics of AL scores were stratified by sex and age categories. Multiple regression analyses were conducted for the first and third follow-up to assess if there were changes in associations between sociodemographic factors and AL.

### Results

Average AL scores of men decreased, while women's AL scores returned to baseline levels after an initial decrease observed at the first follow−up. Stratified analyses of AL scores revealed that women in the younger age cohorts had lower mean AL scores at baseline than men (women: −3.47, 95% CI [−4.24; −2.71] vs. men: −1.13, 95% CI [−1.84; −0.42] at age <55). At the same time, women showed higher mean

**Data availability statement:** Restrictions apply to the availability of these data due to legal restrictions. The data was used under license for the current study, and is therefore not publicly available. The data that support the findings of this study are available from the steering committee upon reasonable request and with permission of the CARLA Study steering committee. Researchers interested in the data can apply for the data by sending an email to [carlastudie@uk-halle.de] or through submitting an application form that is available on the CARLA study website https://webszh. uk-halle.de/carla-studie/. To access the data, a formal application must be submitted with a detailed research proposal consisting of a title, authors, research questions, brief scientific background, list of needed variables, and proposed statistical analyses. All proposals will be reviewed by the CARLA Study steering committee and a final decision on the use of data will be given.

**Funding:** We acknowledge the financial support of the Open Access Publication Fund of the Martin-Luther-University Halle-Wittenberg. The funders did not influence the study design, data collection and analysis, decision to publish, or preparation of the manuscript.

**Competing interests:** The authors have declared that no competing interests exist.

**Abbreviations:** AL, Allostatic Load; BMI, Body Mass Index; CARLA, Cardiovascular risk factors, living and ageing in Halle; CI, Confidence Interval; CRP, C-reactive Protein; HbA$_1$c, Glycated Hemoglobin; HDL, High-density Lipoprotein Cholesterol; LDL, Low-density Lipoprotein Cholesterol; RR, Blood pressure; SES, Socioeconomic Status.

AL scores than men in older age cohorts (women: −0.32, 95% CI [−1.58; 0.95] vs. men: −0.93, 95% CI [−1.99; 0.14] at age 65−<70).

Results of multiple regression models indicated lower AL scores for women (β: −1.21, 95% CI [−1.93, −0.49]). Professional status was associated with lower AL scores for men but not for women (β: −1.06, 95% CI [−2.02, −0.11] for men). Further, physical activity was negatively associated with AL scores for the total study sample and for women (β: −0.54, 95% CI [−0.82, −0.26]) for total sample and β: −0.74, 95% CI [−1.17, −0.32] for women).

## Conclusion

Our results highlight the importance of health awareness and physical activity for overall health, assessed by AL. Distinct AL score changes and sex-specific socioeconomic influences offer insights into sex-related patterns of aging. Further research is needed to understand the underlying mechanisms of socioeconomic influences on stress-related aging processes between sexes.

## Introduction

### Allostatic load as embodiment of stress

Stress is a psychophysiological response of the body to achieve adaption to challenging stressors [1,2]. In this regard, acute stress leads to a physiological adaption to maintain homeostasis if short-term stressors occur. Chronic stress must be distinguished from this physiological response because of the long-lasting character of chronic stressors, which ultimately lead to maladaptive responses of the body [1,3]. This results in the secretion of hormones that affect various organs including the heart, liver, kidney and brain. Several studies have focused on the measurement of chronic stress and investigated multiple biomarkers as a surrogate of chronic stress [3]. One of the most common methods to quantify the physiological response to stressors is Allostatic Load (AL) [4,5]. The concept of AL is based on the concept of allostasis, the organism's attempt to adapt to external influences through a reaction of different organ systems to preserve physiological stability [6]. As such, AL describes the physiological cost of chronic exposure to stress and shows the cumulative "wear and tear" on the body as a result of the constant release of stress hormones [6]. The underlying mechanism consists of the activation of the physiological stress system that initiates the release of measurable mediators. These mediators are used to operationalize AL and can be quantified as an allostatic load score [7,8]. Primary mediators, as part of the first systems to be activated through stress include the neuroendocrine and the immune system [4,5,9].

The main pillars of the neuroendocrine system are the hypothalamic–pituitary–adrenal axis and the sympathetic–adrenal–medullary system, which can be quantified through their metabolites such as cortisol, epinephrine, norepinephrine, dopamine, aldosterone or dehydroepiandrosterone.

Increased primary mediators lead to secondary systemic dysregulation that reflect the cumulative effects in specific tissues or organs [10]. Secondary systemic dysregulation in the metabolic, inflammatory and cardiovascular system are measured by biomarkers for each system [11]. Hence, frequently used biomarkers from the secondary systems such as systolic blood pressure, diastolic blood pressure, waist–hip ratio, high-density lipoprotein cholesterol, total cholesterol, and glycated hemoglobin [8,10] can be considered as the result of compensation for dysregulated stress hormones [4].

Over time, the sustained secretion of stress hormones and subsequent dysregulation in different organ systems can lead to various negative health outcomes, including cardiovascular disease, diabetes, and depression [11]. Thus, AL has been established as a better predictor for mortality [12–15], diabetes mellitus, cardiovascular and other chronic diseases than any single parameter included in the AL score alone [11,13,16].

### Influencing factors of AL

Various determinants of AL have been identified in the literature [17]. In addition to psychosocial [18] and individual behavioral factors [19], studies have shown that socioeconomic factors [20] are associated with AL. In this regard low individual socioeconomic status (SES), defined by low household income [21] and low education [22,23] was associated with high values of AL in previous studies. Other investigations found high socioeconomic position to be related to low AL, which seems to confirm this correlation [24–29].

Further, adverse working conditions and job insecurities seem to have a negative impact on AL [11]. Increased job demands without social support or decision latitude, were associated with higher allostatic load levels in men older than 45 years after controlling for smoking status and age [30]. It has also been shown that vital exhaustion, an effort–reward imbalance and burnout symptoms were associated with higher allostatic levels in female school teachers [31].

In addition, there is considerable amount of evidence on lifestyle habits being associated with AL, such as physical activity [17,32,33], sleep quality [34,35], diet and weight [36–40], alcohol consumption [17,33,41] and smoking habits [42].

In this context, AL has evolved to operationalize the relationship between social inequality, stress and health outcome, and allows to study the relations of social stressors on individuals' health [10,11,20].

### Sex-specific differences in AL

Some studies suggest that women have or develop higher levels of AL than men, particularly in response to certain stressors such as caregiving [43,44].

Overall, the results with regard to sex-specific changes in AL over the life-course are inconsistent [43,45,46]: A cross-sectional study within the US population showed higher AL scores for women than for men and increased differences between sexes at ages over 60 years [43], while a longitudinal cohort study from England found higher AL scores for men and similar changes of AL over time for both sexes [45]. Besides that, the selection of biomarkers for the calculation of AL scores may impact AL score differences between the sexes [43,47]. Previous research also found sex-specific associations between socioeconomic status and AL scores [48]. Since socioeconomic status and other determinants may play a role in an individual's AL, more research is needed to understand the potential sex differences in AL and their change over time [11].

### Age and allostatic load

An increase in AL with age has been shown. This is, among other things, based on the accumulation of stressors over time and can also be explained by an increased dysregulation in organ systems with age [14,45,49]. Beyond that, multiple studies have found an association of AL with frailty [16,50–52] as well as with decline in cognitive and physical functioning [14,15,52–54].

## Measurement of AL

Commonly a combination of multiple physiological markers is used to create an overall AL score [8]. Since there are plenty of studies measuring AL, there is tremendous variation concerning the measurement of AL [8,10]. The most frequent way to calculate AL scores, is to convert each biomarker into a dichotomous variable, attributing one point to the biomarker if it falls in its high-risk range and 0 points if does not. Hereby, high risk is most commonly determined by the highest or (respectively lowest) quartile measured for each biomarker [10,55]. Still, there is a broad agreement that biomarkers from at least three organ systems (cardiovascular, metabolic and immune) should be used to measure AL, and the score should contain biomarkers of both primary and secondary mediators [8]. Further, most authors agree that the use of medication that affect the included organ systems (i.e., anti-hypertensive, lipid-lowering or antidiabetic medication) should be accounted for by various methodic approaches [8].

## Research gap and aim of the study

Despite the existence of longitudinal data on determinants of AL, there is still a limited availability of data that includes multiple measurements of AL throughout an individual's life span [45,46,56]. Only few studies investigated the longitudinal pattern of AL based on repeated measurements in community-based cohorts including over a long period [13]. Moreover, previous research on the relationship between determinants and AL has primarily concentrated on AL as the outcome measured at a single point in time. This approach overlooks associations that may exist for specific durations or exhibit cumulative effects.

This highlights substantial research gaps that can be explored through a longitudinal study of AL over nearly two decades within a cohort study. While cross-sectional studies have reported higher AL scores at older ages, longitudinal investigations of this association within a consistent cohort over extended time periods are lacking [13].

Furthermore, the question arises of how AL changes in different subpopulations within a cohort and whether differences can be identified for these subpopulations. Findings on sex-specific differences in AL have been described but also reveal similar methodological limitations. Studies investigating the longitudinal pattern of AL by sex show inconsistent results and often focus on shorter observation periods [45,46]. Further, associations between age and AL within a cross-sectional framework do not provide insights into generation- or age-cohort-specific relationships.

Within the scope of a longitudinal investigation of AL, we can explore potential changes of AL over time across sex and various age cohorts, thereby advancing the current state of research.

To investigate longitudinal patterns of AL, we focused especially on sex- and age-specific relationships of socioeconomic and individual behavioral factors with AL in the CARLA-cohort over a period of almost 20 years. Due to the comprehensive examinations at multiple points in time and the large number of parameters, data of the CARLA study provides the chance to investigate longitudinal changes of AL levels. Hereby, this work pursues the following questions:

1. Do mean AL scores change in the CARLA-cohort over the study period for CARLA?

2. Do mean AL scores change differently for men and women and within age-groups?

3. Do the associations between socioeconomic and behavioral factors with AL scores change over time?

## Methods

### The study population and data source

This article is based on an analysis of data from the CARLA study, a prospective cohort study that investigates risk factors for cardiovascular diseases and determinants of successful ageing. Detailed description of the CARLA study was published elsewhere [57–59]. The CARLA study has been approved by the responsible ethics committee (registration

number: 164/12.10.05/1). For this analysis, no additional ethics clearance was required. In brief, the cohort was recruited using the population registry of the city of Halle. The baseline examination took place July 2002 through January 2005, followed by the first follow-up examination between March 2007 and March 2010. The second follow-up took place between January and October 2013, and the third between October 2020 and March 2022 [59]. The data was accessed for this research in November 2022.

The participation proportion was 64.1% (68.6% for men and 59.5% for women) and the retention proportions were 81%, 77%, and 27% of the original cohort through follow-up examinations 1–3, respectively (Fig 1). The study population showed a high prevalence of hypertension (74%), type 2 diabetes (15%) and a high average Body Mass Index (BMI) (28 kg/m2) at baseline [58]. As the second follow-up does not contain laboratory parameters that are needed to calculate AL, we decided to utilize data of the baseline (CARLA-0), first (CARLA-1) and third follow-up (CARLA-3) examinations for this analysis. Given the loss of participants over time, we included only those who participated in each follow-up. Since this approach can potentially lead to selection bias, we conducted a descriptive and analytical sensitivity analysis. We present descriptive summaries of the entire study sample (n = 1779) compared to the subsample (n = 473) in the supplement (S1 Table). Further, we tested differences between these samples using t−test (continuous variables) or χ2 test (categorical variables). We report the mean difference and 95% confidence intervals: Participants who attended at all examinations were found to be younger at baseline (−6.6 years, 95% CI: [−7.4; −5.8]), had higher income, educational level and professional status, reported fewer pack−years of tobacco use (7.5 versus 10.4; 95% CI: [−4.19; −1.69]), engaged in more hours of physical activity per week (0.95 versus 0.75, 95% CI: [0.05; 0.35]), and had a lower proportion of medication use (e.g., antihypertensive medication, 0.38 versus 0.55, χ2= 77.201, p−value: <2.2e−16). Solely, consumption of alcohol (g per day, 12.0 versus 11.4, 95% CI: [−0.89; 2.19]) was not found to be different between the subgroup and the whole sample. To assess the impact of missing at random on our results, we conducted the linear regression models for the entire population of CARLA−0 (n = 1779) and CARLA−1 (n = 1436) and compared these with the results of the regression models for the subpopulation of CARLA−0 (n = 473) and CARLA−1 (n = 473). Since the effect estimates

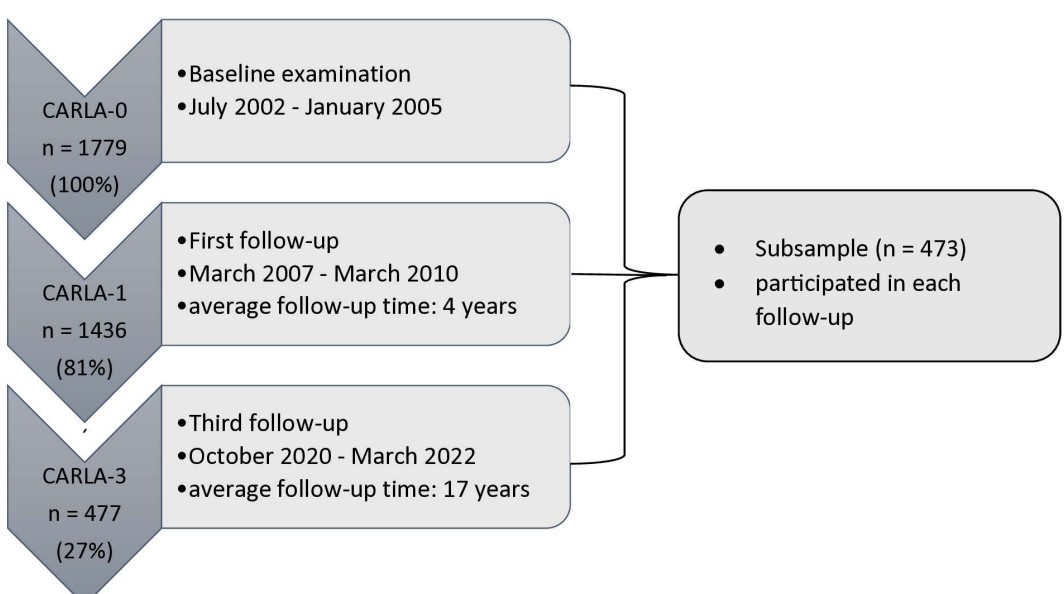

**Fig 1.  Flow chart: Retention proportion and subsample of the CARLA cohort.**

differed only nominally, but not in their direction and significance, we decided to proceed with the analysis of the subpopulation to ensure data comparability of the study population over time without distorting the estimated effects.

## Allostatic load measurements and allostatic load score

We calculated an AL score to measure biological dysregulation at baseline and follow-up examinations 1 and 3 by gathering data on metabolic, immune, cardiovascular and anthropometric measures. Anthropometric indicators included waist circumference and body-mass index (BMI). As metabolic measurements, we used triglycerides (TG), low-density lipoprotein cholesterol (LDL), high-density lipoprotein cholesterol (HDL), and glycated hemoglobin (HbA$_1$c). Cardiovascular measures included systolic and diastolic blood pressure, while c-reactive protein (CRP) was used for immune measurement. As non-fasting venous blood samples were used at each examination time point (59), we highlight the intraindividual measurement variability of the metabolic biomarkers. While HDL and LDL exhibit intraindividual variability of 4–8% (for HDL) and 5–10% (for LDL), respectively, triglycerides may show higher variability of 20–30%, depending on factors such as the time of measurement and diet [60–63]. Due to the availability of single measurements at each follow-up, a methodological consideration of these intraindividual variabilities is not possible. Blood pressure was measured three times following an initial 5-minute rest period, with a 3-minute break between each measurement. The final blood pressure values are the average of the last two readings [59].

Due to missing biomarker data for a few participants, resulting in missing AL scores, we imputed the absent biomarkers using the mice-function in R studio [64]. Since nine biomarkers were measured at baseline and two follow-ups, there are 27 measurements. Of these 27 variables, eight variables were complete for all study participants. For 13 of the variables, there were a maximum of five missing data points. For another six variables, there were up to 33 missing data points. A detailed breakdown of the missing data points can be found in the supplement (S2 Table).

In order to summarize these measurements, we first calculated standardized z-scores for each of the nine indicators. We then summed standardized z-scores of all parameters for each individual to create an AL score at baseline and follow-up examinations 1 and 3. The approach of using z-scores to calculate AL scores has become established through its utilization in numerous studies [4,23,65,66] and provides the opportunity to account for the continuity of the measured variables [65]. To enable a comparison of AL over time, we calculated standardized z-scores using cut points for higher risk derived from clinical guidelines. We set these clinical cut points as zero to indicate where each individual's value placed them (in SD units) relative to these cut points. The clinically established cut points used in our analysis are shown in Table 1. A high standardized score implies a higher risk except for high-density lipoprotein. To reflect lower values of high-density lipoprotein corresponding to higher biological risk and to create a concurring score, we multiplied the standardized score of high-density lipoprotein by −1 before calculating the AL score. We decided to calculate the AL score by adding up z-scores of each parameter as it allows the weight of each biomarker to be equal depending on its deviation from the cut-point's mean. Despite our reasoned decision to use z-scores, we calculated two alternative sum score for AL to check if differences appear based on the method of operationalization. First, we assigned one point if the biomarker fell within high-risk categories based on clinical guidelines and summed them up to an overall sum score. Following previous studies, we decided to add half a point within a second alternative score for participants taking medication and who had low scores for blood pressure, glycohemoglobin, or lipids [10,67]. We found no difference in trends of AL comparing all three AL scores. Results of alternatively calculated AL scores can be found in the supplement (S3 Table). Taking this into consideration, we assume that the approach of using z-scores is appropriate in the context of our cohort.

## Other variables

In line with Lampert et al. (2013), we considered three variables as a surrogates to measure socioeconomic status [71]. The household net equivalent income (accounting for the number of persons in the household) at baseline was classified in seven categories (< 500€ = 1, 500 −< 750€ = 2, 750 −< 1000€ = 3, 1000 −< 1500€ = 4, 1500 −< 2000€ = 5, 2000 −< 2500€

**Table 1. Biomarkers used to calculate AL score for comparison over time** *(Corresponding publications are indicated in brackets).*

| Biomarker | Clinical cut-point |
|---|---|
| **Waist circumference** | 94 cm for men and 80 cm for women [68] |
| **Body Mass Index (BMI)** | 30 kg/m² [68] |
| **CRP**[a] | 3 mg/l [69] |
| **Triglycerides** | 1,7 mmol/l (150 mg/dl) [70] |
| **LDL**[b] | 3,0 mmol/l (115mg/dl) [70] |
| **HDL**[c] | 1,0 mmol/l (40 mg/dl) for men<br>1,2 mmol/l (48mg/dl) for women [70] |
| **HbA1$_c$**[d] | 7% (53 mmol/l) [70] |
| **RR$_{sys}$**[e] | 140 mmHg [70] |
| **RR$_{dia}$**[f] | 90 mmHg [70] |

[a]*c-reactive protein.*

[b]*low-density lipoprotein.*

[c]*high-density lipoprotein.*

[d]*glycated hemoglobin,*

[e]*systolic blood pressure,*

[f]*diastolic blood pressure*

= 6,>= 2500€ = 7) [57]. In order to differentiate between the impact of lower and higher net equivalent incomes on AL, we categorized net equivalent income in low and high income based on the median net equivalent income in east Germany in 2003 [72]. Participants having a net equivalent income lower than 1500 € were allocated to the low−income category.

Education was measured by the highest attained school education (no formal education = 1, elementary school (Hauptschule) = 2, secondary school (Realschule in FRG) = 3, secondary school (POS in GDR) = 4, qualified for university of applied sciences (Fachhochschulreife) = 5, A−levels/university entrance qualification (Abitur) = 6). Similarly with the procedure described above, we categorized into low educational level (secondary school or lower) and high educational level (secondary school (POS in GDR) or higher).

Professional status was measured in 7 categories (unskilled worker= 1, semiskilled, skilled worker, farmer = 2, foreman, assisting family, simple employee, lower civil servant = 3, employee foreman, qualified employee, middle civil servant = 4, self-employed <10 employees = 5, high qualified employee, higher civil servant, self-employed academic/professional =6, Employee with complex leadership, senior civil servant, self-employed>=10 employees = 7). We considered categories 1–4 as low professional level and 5–7 as high professional level.

To evaluate the influence of lifestyle variables, we took smoking status into account (current = 1, ex = 2, never = 3) and added cumulative pack years into the multiple regression model. Further, we included alcohol consumption (g/day) and physical activity (sport h/week) in our analysis. AL is also influenced by other social determinants of health like marital status and number of children. Therefore, we included marital status in our analysis coded as single=1, married=2, divorced=3 and widowed=4 and the number of biological children.

For medication use we took three measures into account: use of hypertension medication, use of lipid lowering medication and use of antidiabetic medication.

## Statistical analysis

In a first step, we performed descriptive statistics to investigate AL in the study cohort at each of the three time points. Since our study is an exploratory analysis, we report means and confidence intervals instead of the p-value to avoid data dredging or p-hacking [73,74].

In a second step, we performed multiple linear regression to investigate the association of individual socioeconomic variables (income, education and professional status) and behavioral factors (use of tobacco, alcohol consumption, physical activity, marital status, number of children) with AL scores. As this study specifically aims to explore the association between these independent variables and AL scores within our cohort at different follow-ups, we rely on the baseline values of the predictors.

We adjusted for confounders such as age, sex, and medication use at the different time points. We emphasize that we have explicitly decided against adjusting for the baseline level of AL based on the following four reasons:

1. Adjusting for the baseline value of a dependent variable carries the risk of collider bias. In this case, one controls for a variable that may be influenced both by the independent variables and by unobserved factors [75]. In the present model, this could lead to an overestimation of the associations between the independent variables and AL.

2. As we are analyzing a cohort with a high average age, where some independent variables (e.g., educational attainment) exerted their effects long before the baseline examination, the AL at baseline partially reflects the associations with these independent variables [76]. Adjusting for AL at baseline would thus underestimate these associations.

3. Adjusting for the AL at baseline would measure only the associations between the independent variables and AL that go beyond the effect of AL at baseline [77]. This would capture changes in AL, which is not the research focus of the present study.

4. A correlation between the baseline value of a dependent variable and subsequent measurements is possible and could lead to overfitting, thereby impairing the predictive power and robustness of the model [78].

We applied two different models: For the first model, we explored associations between the different factors at baseline and AL scores after a mean follow-up of 4 years at the first follow-up (Model 1). For the second model, we aimed to investigate associations of these factors with AL scores at the third follow-up after a mean follow-up period of almost 17 years. Additionally, we applied each model separately for men and women. The statistical analyses were performed using the RStudio software environment.

## Results

### Demographic characteristics at baseline

The study sample consisted of 473 participants, of which 245 were men (52%) and 228 were women (48%) aged 45–80 years. The mean age at baseline examination was 58 years.

Considerable differences in baseline characteristics between men and women existed in their marital status, alcohol consumption, use of nicotine and cigarette pack-years: 83% of men but only 71% of women were married at baseline. In contrast, the percentage of widowed women was noticeably higher (10% of women vs. 2% of men), consistent with the typical age difference in marriages.

Sex differences were also evident in the lifestyle factors: On average, men consumed 3.6 times more alcohol than women (18.4 g/day vs. 5.2 g/day) and reported an average of 10.8 pack-years of tobacco, while women reported 3.9 pack-years. Also, only 36% of all men, but 66% of all women claimed to have never smoked. In addition, the proportion of current smokers was higher among men than among women (22% vs. 15%). Physical activity and the intake of medications were not found to be different between men and women. Details concerning the demographic, socioeconomic and behavioral variables at baseline are shown in Table 2.

### Changes of socioeconomic and behavioral characteristics over time

In our cohort, net equivalent income and professional status did not change noticeably from baseline to the first follow-up. Given that the youngest individual included in the cohort at baseline was 44 years old and the average age was 58 years,

Table 2. Demographic, socioeconomic and behavioral properties of the study population at baseline.

| Characteristics | | Men n=245 | Women n=238 | All n=473 |
|---|---|---|---|---|
| mean age (years) | | 57.9 | 57.6 | **57.7** |
| income | low[a] | 50.2% | 53.5% | **51.8%** |
| | high | 49.8% | 46.5% | **48.2%** |
| education | low[b] | 17.1% | 19.7% | **18.3%** |
| | high | 82.9% | 80.3% | **81.6%** |
| profession | low[c] | 49.8% | 77.2% | **63.0%** |
| | high | 52.2% | 22.8% | **37.0%** |
| number of children (mean) | | 1.73 | 1.56 | **1.65** |
| marital status | divorced | 11.0% | 13.4% | **12.2%** |
| | married | 82.9% | 71.1% | **77.1%** |
| | single | 4.1% | 5.6% | **4.8%** |
| | widowed | 2.0% | 9.9% | **5.9%** |
| use of nicotin | current smoker | 21.6% | 14.5% | **18.2%** |
| | ex smoker | 42.9% | 19.7% | **31.7%** |
| | never smoked | 35.5% | 65.8% | **50.1%** |
| alcohol consumption | g/day (mean) | 18.4 | 5.2 | **12.0** |
| pack-years of tabacco | years (mean) | 10.8 | 3.9 | **7.5** |
| sport | hours/week (mean) | 0.95 | 0.94 | **0.95** |
| antihypertensive medication | yes | 34.3% | 42.1% | **38.1%** |
| lipid lowering medication | yes | 10.6% | 7.9% | **9.3%** |
| antidiabetic medication | yes | 4.1% | 4.8% | **4.4%** |
| | | | | |

[a]Net equivalent income of the household< 1500€.

[b]secondary school or lower.

[c]unskilled worker, semiskilled, skilled worker, farmer, foreman, assisting family, simple employee or lower civil servant, employee foreman, qualified employee, middle civil servant

we can infer that there were no changes in school education during the follow-up period. As a result, we treated these variables as time-invariant and chose not to display them.

The proportion of participants taking antihypertensive drugs increased by 19 percentage points for women and 10 percentage points for men in the first follow-up compared to the baseline examination (S4 Table). Similar developments were seen for antidiabetic and lipid-lowering medication.

Further, lifestyle factors changed over time. At the time of the first follow−up, the proportion of current and occasional smokers decreased for both sexes compared to the proportion of current smokers at baseline examination (−5.2 percentage points for men and −3.4 percentage points for women). Further, hours of physical activity per week increased by 0.75 hours for men and 0.95 hours for women.

## AL at baseline and in the follow-ups

Average AL scores decreased over time from −1.66 (95% CI: −1.98; −1.34) at baseline to −2.30 (95% CI: −2.57; −2.02) at the third follow−up. That means, in the 20 years between the examination, average AL decreased by about halve a standard deviation.

Men, participants with low education or professional level and participants who took antidiabetic or antihypertensive medication showed higher mean AL scores compared to the whole sub−cohort at all examinations. Further, information on mean AL scores for baseline and follow-up examinations stratified by sex and baseline characteristics are shown in Table 3.

## Stratification by sex and age cohorts

At all examination times, women had a lower AL score than men. The decrease of AL scores for the whole sample was evident for men, but not for women. Thus, the AL score for men decreased from −0.90 (95% CI: −1.32; −0.48) at baseline to −2.08 (95% CI: −2.47; −1.70) at the third follow−up. In contrast, the mean AL score for women decreased from −2.47 (95% CI: −2.93; −2.01) at baseline to −2.69 (95%CI: −3.16; −2.22) at the first follow−up, but increased at the last follow−up to a level comparable to their baseline AL score (−2.53 (95% CI: −2.93; −2.12)).

The following boxplot (Fig 2) illustrates a stratified analysis of AL scores by sex and age. For parsimony and for illustrational purpose, we dichotomized our cohort in two categories (over vs. under 65 years). More detailed information about stratified analysis (including confidence intervals and all age cohorts) can be found in the supplement (S5 and S6 Tables). AL score of men in the younger age cohorts (<65 years) were higher than the AL score of women at all examination times and their AL score increased up to the age cohort of <65 years. During the examination period the AL score of each age cohort gradually decreased for men. Although women had lower scores than men in the age cohorts of <65 years, they overtook them at age cohorts over 65 years. Since confidence intervals for some age cohorts overlap, those results need to be interpreted cautiously.

## Regression analyses

In the total analyzed sample, professional status was associated with AL scores in all models, whereby low status was connected to higher AL. When calculating separate regression models for each sex, this association persisted for men.

There was no association between education or income and AL scores in both models. In regards to lifestyle behaviors, hours of physical activity per week was negatively associated with AL scores at the both follow-ups in the total sample and for women. Further, we found an inverse association between age (in years) and AL scores at the first follow-up for women, but not for men. Interestingly, female sex was associated with lower AL scores at both follow-ups indicating sex differences of AL scores to decrease within an aging cohort. No associations were found for smoking status, consumption of alcohol and marital status. Results of regressions analyses are shown below (Table 4 and 5).

## Discussion

We found a decrease of average AL scores in our analyzed sample over the follow-up period. This trend was evident for men, but only partially for women. Our stratified statistical analysis for men and women revealed differences in the association of the professional status and AL scores. In regards to behavioral variables, only physical activity was associated with AL scores, while smoking and alcohol consumption did not exhibit relations with AL scores.

In reference to the decreased AL scores over time, it is important to note that some systematic changes occurred during the follow-up period. In particular, participants with systolic blood pressure measurements exceeding >140/90 mmHg at baseline were instructed to inform their physician. In the follow-up, we observed a substantially higher proportion of participants with hypertensive medication. Thus, the observed improvement in controlled blood pressure could have resulted from this intervention, leading to advantageous AL scores, particularly for men [79]. Similarly, increased attention towards health problems resulting from the baseline examination could have led to the increase of physical activity levels and reduced percentage of current smokers that we have seen at the first follow-up. The potential effects of these shifting lifestyles cannot be conclusively evaluated due to low retention and the risk of selection bias. Most previous research has shown increasing

**Table 3. AL scores at baseline and follow-ups.**

| | | AL-score baseline (n=473) Mean [95% CI] | AL-score first follow-up (n=473) Mean [95% CI] | AL-score third follow-up (n=473) Mean [95% CI] |
|---|---|---|---|---|
| | total | −1.66 [−1.98; −1.34] | −2.14 [−2.45; −1.83] | −2.30 [−2.57; −2.02] |
| **sex*** | men (n = 245) | −0.90 [−1.32; −0.48] | −1.63 [−2.04; −1.22] | −2.08 [−2.47; −1.70] |
| | women (n = 228) | −2.47 [−2.93; −2.01] | −2.69 [−3.16; −2.22] | −2.53 [−2.93; −2.12] |
| **income*** | low income [a] | −1.51 [−1.95; −1.07] | −2.02 [−2.46; −1,59] | −2,19 [−2.57; −1.81] |
| | high income | −1.81 [−2.28; −1.35] | −2.27 [−2.72; −1.82] | −2,41 [−2.81; −2.00] |
| **education*** | low education [b] | −0.96 [−1.69; −0.22] | −1.29 [−2.02; −0.56] | −2.01 [−2.62; −1.39] |
| | high education | −1.82 [−2.17; −1.46] | −2.33 [−2.67; −1.99] | −2.36 [−2.67; −2.05] |
| **profession*** | low profession [c] | −1.53 [−1.93; −1.12] | −1.85 [−2.25; −1.14] | −2.06 [−2.41; −1.71] |
| | high profession | −1.88 [−2.40; −1.37] | −2.64 [−3.12; −2.15] | −2.70 [−3.15; −2.26] |
| **smoking status*** | current smoker | −1.61 [−2.53; −0.69] | −2.37 [−3.28; −1.46] | −2.28 [−3.03; −1.55] |
| | ex smoker | −1.06 [−1.58; −0.54] | −1.75 [−2.27; −1.23] | −2.22 [−2.70; −1.74] |
| | never smoked | −2.05 [−2.48; −1.62] | −2.31 [−2.72; −1.90] | −2.35 [−2.73; −1.97] |
| **marital status*** | single | −2.08 [−3.55; −0.61] | −2.93 [−4.43; −1.43] | −2.72 [−3.99; −1.44] |
| | married | −1.49 [−1.86; −1.23] | −2.01 [−2.37; −1.66] | −2.17 [−2.49; −1.86] |
| | divorced | −2.34 [−3.14; −1.54] | −2.82 [−3.78; −1.87] | −2.78 [−3.62; −1.93] |
| | widowed | −2.06 [−3.49; −0.63] | −1.72 [−2.81; −0.62] | −2.61 [−3.73; −1.49] |
| **use of medication*** | antihypertensive medication | −0.49 [−0.98; 0.01] | −1.22 [−1.69; −0.74] | −1.89 [−2.35; −1.43] |
| | lipidlowering medication | −1.00 [−2.04; 0.03] | −1.07 [−2.20; −0.04] | −1.60 [−2.59; −0.61] |
| | antidiabetic medication | 0.75 [−1.01; 2.51] | 0.97 [−0.90; 2.85] | −1.43 [−2.66; −0.20] |

[a]Net equivalent income < 1500€

[b]secondary school or lower

[c]unskilled worker, semiskilled, skilled worker, farmer, foreman, assisting family, simple employee or lower civ. Servant, employee foreman, qualified employee, middle civil servant

*covariates assessed at baseline

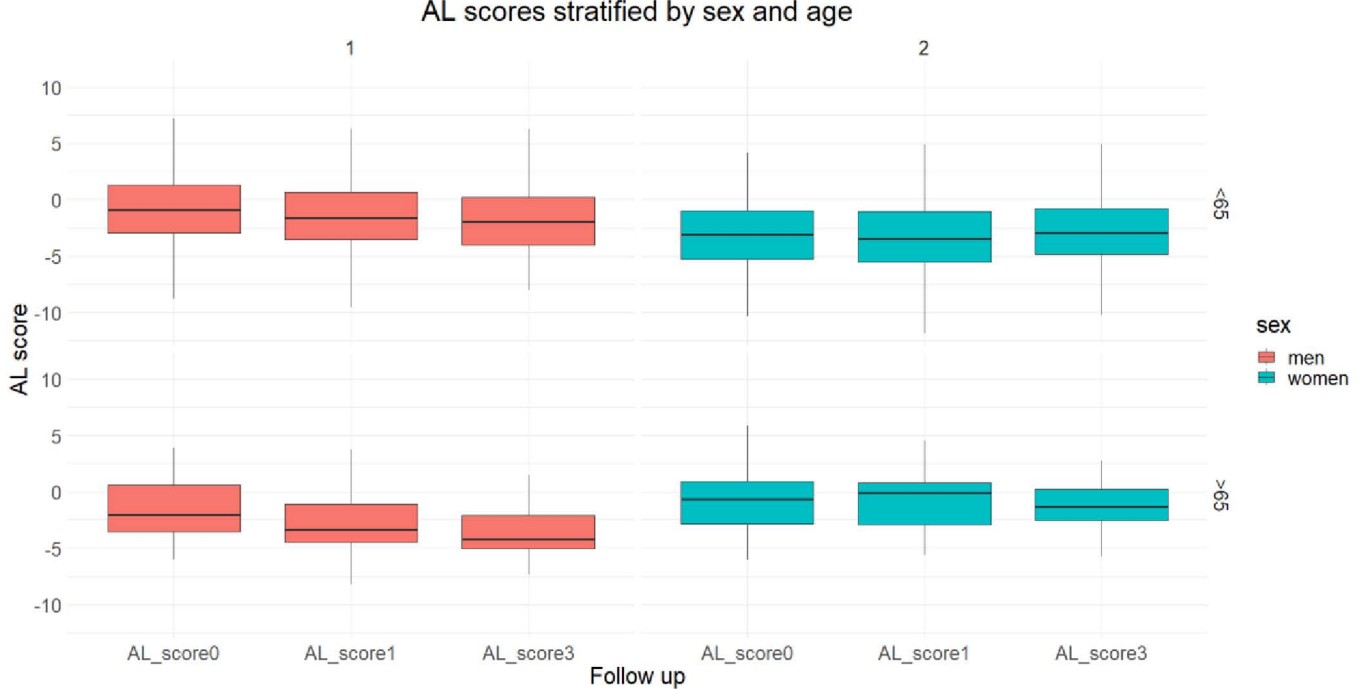

**Fig 2. AL scores stratified by sex and age cohorts at different examinations in the CARLA study.**

trends of AL scores for aging cohorts [14,45,56]. This observation of increasing AL scores with time is also biologically plausible, since the chronic maladaptation to stress – which is measured by AL – accumulates over time [6]. Meanwhile, in our sample the accumulation of stressors over time seems to have been countered by changes in lifestyle.

We found women to have a lower AL score at all examinations, but distinct patterns of AL scores over the time were evident when comparing sexes. In a stratified analysis for sex and age cohorts, mean AL scores of men in the younger age cohorts (<65 years) were higher than the AL score of women at all examination times, but lower in the older age cohorts, suggesting different pathways for men and women. These observations are in line with a study by Yang and Kozloski, who found women to have a lower AL score than men, but with a steep postmenopausal increase that persisted after adjusting for other covariates, causing women to have higher AL scores than men approximately 10 years after menopause [43].

Despite the fact that women overtook men in AL score at older ages in our cohort compared to investigations by Yang and Kozloski, we found similar developments, which we suspect in sex-specific hormonal differences. Yang and Kozloski described a lag of 5–10 years between the mean age of menopause and the increase of female AL scores. They explained this lag due to the wide use of estrogen replacement therapies among U.S. women and the latency of physiological dysregulation [43]. We observed a lag of almost 15 years after mean age of menopause in our cohort (49 years) that might be explained by the inaccuracy that arises through the stratification in age groups of 5 years. Overall, these sex-specific results are consistent with previous research [43,45,46].

Moreover, we found that a low professional level was associated with low AL scores at the last follow-up in the whole sample. In the analysis performed separately for the sexes, this relation did was more visible for men. In contrast with our findings, Dowd et al. found adolescent socioeconomic status independently inversely related to AL only for women after adjustment for smoking, snuff use, alcohol and physical inactivity [48]. Another study found lower occupational status strongly negatively correlated with AL, supporting our results [80].

**Table 4. Factors associated with AL score (multivariable regression analyses for the first follow up).**

| | Model First Follow up (Total) β 95%CI | Model First Follow up (men) β 95%CI | Model First Follow up (women) β 95%CI |
|---|---|---|---|
| (Intercept) | −1.39 | −3.33 | −2.95 |
| | [−3.66, 0.88] | [−6.73, 0.06] | [−5.54, −0.36] |
| high income vs. low[a] | 0.44 | 0.56 | 0.15 |
| | [−0.20, 1.08] | [−0.30, 1.43] | [−0.84, 1.14] |
| high education vs. low[b] | −0.14 | −0.37 | 0.06 |
| | [−1.02, 0.73] | [−1.72, 0.98] | [−1.20, 1.33] |
| high profession[c] vs. low[c] | −1.20 | −1.06 | −0.99 |
| | [−1.88, −0.52] | [−2.02, −0.11] | [−2.09, 0.11] |
| ex smoker[d] | 0.27 | 0.03 | 0.18 |
| | [−0.72, 1.25] | [−1.26, 1.32] | [−1.64, 1.99] |
| never smoker[d] | 0.27 | −0.36 | −0.12 |
| | [−0.90, 1.44] | [−2.03, 1.31] | [−2.30, 2.05] |
| married[e] | 0.28 | 1.59 | −0.57 |
| | [−1.21, 1.77] | [−1.24, 4.43] | [−2.42, 1.28] |
| divorced[e] | −0.09 | 1.25 | −1.23 |
| | [−1.78, 1.60] | [−1.91, 4.41] | [−3.25, 0.79] |
| widowed[e] | 0.60 | 1.66 | −0.56 |
| | [−1.07, 2.27] | [−1.23, 4.55] | [−2.53, 1.42] |
| packyears[f] | 0.04 | 0.16 | −0.48 |
| | [−0.46, 0.54] | [−0.50, 0.82] | [−1.22, 0.25] |
| consumption of alcohol[f] | 0.11 | 0.01 | 0.09 |
| | [−0.27, 0.48] | [−0.43, 0.46] | [−0.41, 0.60] |
| sport[f] | −0.54 | −0.31 | −0.74 |
| | [−0.82, −0.26] | [−0.72, 0.09] | [−1.17, −0.32] |
| number of children[f] | 0.26 | 0.16 | 0.33 |
| | [−0.08, 0.61] | [−0.38, 0.69] | [−0.10, 0.77] |
| age[f] | 0.19 | −0.29 | 0.80 |
| | [−0.07, 0.59] | [−0.77, 0.19] | [0.31, 1.28] |
| women vs. men | −1.21 | | |
| | [−1.93, −0.49] | | |
| antidiabetic mediaction vs. no | 1.54 | 1.45 | 1.75 |
| | [0.92, 2.16] | [0.56, 2.33] | [0.85, 2.66] |
| lipidlowering medication vs. no | −0.34 | −0.41 | −0.05 |
| | [−1.15, 0.47] | [−1.56, 0.75] | [−1.27, 1.17] |
| antihypertensive medication vs. no | 2.55 | 2.75 | 2.11 |
| | [0.71, 4.38] | [0.12, 5.38] | [−0.49, 7.70] |
| N | 473 | 245 | 228 |
| $R^2$ | 0.21 | 0.17 | 0.31 |

Standard errors are heteroskedasticity robust.

[a]Net equivalent income per person < 1500€,

[b]secondary school or lower,

[c]unskilled worker, semiskilled, skilled worker, farmer, foreman, assisting family, simple employee or lower civil servant, employee foreman, qualified employee, middle civil servant

[d]reference: current smoker,

[e]reference: single,

[f]Continuous predictors are scaled by 1 s.d.

**Table 5. Factors associated with AL score (multivariable regression analyses for the third follow-up).**

| | Model Second follow-up (total) β [95%CI] | Model Second follow-up (men) β [95%CI] | Model Second follow-up (women) β [95%CI] |
|---|---|---|---|
| (Intercept) | − 2.11 | −1.15 | −4.79 |
| | [−4.04, −0.17] | [−3.95, 1.64] | [−7.67, −1.91] |
| high income vs. low[a] | 0.15 | −0.59 | 0.73 |
| | [−0.43, 0.72] | [−1.42, 0.25] | [−0.11, 1.57] |
| high education vs. low[b] | −0.20 | 0.42 | −0.67 |
| | [−0.95, 0.55] | [−0.84, 1.68] | [−1.72, 0.39] |
| high profession[c] vs. low[c] | −0.83 | −0.72 | −0.58 |
| | [−1.44, −0.23] | [−1.61, 0.17] | [−1.52, 0.35] |
| ex smoker[d] | −0.06 | −0.84 | 0.82 |
| | [−0.93, 0.81] | [−1.91, 0.24] | [−0.99, 2.63] |
| never smoker[d] | 0.16 | −1.04 | 1.08 |
| | [−0.86, 1.17] | [−2.32, 0.25] | [−1.05, 3.21] |
| married[e] | 0.44 | −0.47 | 0.97 |
| | [−0.90, 1.79] | [−2.79, 1.86] | [−0.97, 2.92] |
| divorced[e] | 0.05 | −0.84 | 0.29 |
| | [−1.45, 1.55] | [−3.39, 1.71] | [−1.83, 2.42] |
| widowed[e] | 0.12 | −1.89 | 0.81 |
| | [−1.62, 1.86] | [−6.12, 2.34] | [−1.45, 3.06] |
| packyears[f] | 0.06 | −0.04 | 0.03 |
| | [−0.34, 0.46] | [−0.53, 0.46] | [−0.71, 0.78] |
| consumption of alcohol[f] | 0.11 | 0.06 | −0.04 |
| | [−0.18, 0.40] | [−0.30 0.43] | [−0.55, 0.46] |
| sport[f] | −0.40 | −0.08 | −0.63 |
| | [−0.66, −0.15] | [−0.46, 0.30] | [−0.99, −0.26] |
| number of children[f] | 0.15 | 0.19 | 0.15 |
| | [−0.13, 0.44] | [−0.23, 0.61] | [−0.26, 0.55] |
| age[f] | −0.14 | −0.37 | 0.16 |
| | [−0.44, 0.17] | [−0.85, 0.11] | [−0.30, 0.62] |
| women vs. men | −0.69 | | |
| | [−1.30, −0.08] | | |
| antidiabetic medication vs. no | 1.79 | 2.12 | 1.42 |
| | [0.90, 2.68] | [0.92, 3.31] | [0.07, 2.77] |
| lipidlowering medication vs. no | −1.76 | −1.81 | −1.57 |
| | [−2.35, −1.17] | [−2.63, −0.99] | [−2.46, −0.68] |
| antihypertensive medication vs. no | 1.61 | 1.41 | 1.61 |
| | [1.02, 2.20] | [0.59, 3.31] | [0.66, 2.55] |
| N | 473 | 245 | 228 |
| R² | 0.18 | 0.24 | 0.24 |

Standard errors are heteroskedasticity robust.

[a]Net equivalent income per person < 1500€,

[b]secondary school or lower,

[c]unskilled worker, semiskilled, skilled worker, farmer, foreman, assisting family, simple employee or lower civil servant, employee foreman, qualified employee, middle civil servant

[d]reference: current smoker,

[e]reference: divorced,

[f]Continuous predictors are scaled by 1 s.d.

The potential difference in the relevance of the professional status for AL between men and women does not align with the literature beyond socioeconomic status: Further environmental stressors have also been found to be stronger predictors of AL for women than for men [81]. Interestingly, Gustafsson et al. found low socioeconomic status at 43 years related to higher AL. After adjustment for tobacco, alcohol use and physical activity, these effects were attenuated in men but not women [9]. Overall, literature indicates that the effect of low socioeconomic status among men is mediated by known behavioral risk factors, while for women the association between socioeconomic status and health is weaker and the pathway unclear.

Regarding behavioral variables, we found hours of physical activity per week negatively correlated with AL scores at both follow-up's for women and the total sample. These findings are partially consistent with results of Gay et al., who found a significant relation between high levels of activity and lower total allostatic load compared to sedentary participants after adjustment for age, sex and education [32]. Although they found a negative association between levels of activity and AL, they did not find significant influences on AL scores across moderate and low active groups. In reference to the relatively low duration of physical activity in our cohort, the effect on AL score even after a long follow-up period remains and raises the question about the dose-response-relationship between physical activity and AL. Even though our results are congruent with previous studies [17,32,33,46,82], there is a need for further investigations to study the dose-response-relationship between physical activity and AL.

In our analysis, alcohol consumption was not associated with AL scores. Other studies found a positive relation between alcohol consumption and AL [16,33,41]. Petrovic et al. found heavy drinking men, and women abstaining from alcohol to have a higher AL than moderate drinkers [17]. In line with findings of Hawkley et al., who showed that the use of current abstainers as the reference group leads to systematic bias, our results may be interpreted as a result of misclassification [15] since abstainers, who may already suffer the consequences of previous alcohol consumption are measured the same as non-drinkers. This relationship has been described as J-shaped and may have occurred in our study [83]. Similar connections to those described and a sample size that is comparatively limited could have resulted in smoking status having no impact on AL scores.

### Strength and limitations of the study

The main strength of the study is that it is based on the extensive set of clinical, laboratory, and lifestyle information over a long follow-up period in a population-based study sample. This allowed us to compare the AL scores of the study cohort across three cross-sections, stratified by sex and age cohorts.

Further, the data provided the chance to investigate determinants of AL in a longitudinal cohort study, with a mean follow-up period of 17 years and conduct sex-specific analyses.

Nonetheless, our results should be interpreted in light of their limitations. The persons included in our analyses are those who were still alive and willing to participate up to 20 years after baseline examination. Although interviews were performed by trained study nurses, sociodemographic and behavioral data relies on self-reported answers, which may have led to a recall bias. We suspected the chosen method of operationalizing of AL to partially explain the contrary results of the development of AL over the follow-up period as we were not able to incorporate medication use in our AL score. Nonetheless, Geronimus et al. 2006 stated that this approach is sensible within cohorts of older ages in which prevalence of chronic diseases are high [53]. In this regard, we highlight the high prevalence of hypertension, type 2 diabetes and a high average BMI (28 kg/m2) at baseline [58]. Since there is no consensus about the operationalization of AL, our procedure of adding up calculated z-scores has some advantages over other approaches, especially considering the multiple measurements of AL scores. To address these requirements, we chose to calculate z-scores for each biomarker and added up each biomarker's z-score to an overall AL score. This approach might also be more precise to estimate associations of different factors with AL scores and allows to compare AL scores within the same cohort over the observation period. However, alternative methods to calculate AL scores led to similar results.

## Conclusion

We examined changes in AL scores over time and explored the potential determinants influencing these scores. The results demonstrated a decrease in AL scores from baseline examination to the third follow-up for men and a plateau for women. This may be partially explained by an improvement in healthcare and changed lifestyle behaviors. Interestingly, the changes in AL scores were distinct for men and women. Notably, women exhibited lower mean AL scores than men in the younger age cohorts, but this trend reversed 15 years after the mean age of menopausal transition, with women surpassing men in the same age cohorts. These results imply that men and women experience distinct aging patterns, as indicated by the varying levels of physiological burden measured by AL.

Furthermore, socioeconomic variables were found to impact AL scores differently for men and women. Specifically, low professional status was associated with higher AL scores for men but not for women. These findings suggest that time-invariant socioeconomic factors, such as professional status, may exert a long-lasting influence on health, but the importance of specific socioeconomic factors varies between sexes.

The study also examined the relationship between lifestyle factors and AL scores. Hours of physical activity per week exhibited a negative correlation with AL scores, indicating that higher levels of physical activity may contribute to lower AL scores especially showing an impact on women. However, the precise amount of physical activity required to achieve this effect needs further investigation. In contrast, alcohol consumption showed a no correlation with AL scores, potentially influenced by misclassification and an abstainer bias.

In summary, this study provides valuable insights into the dynamics of AL scores over time, highlighting the role of sex, socioeconomic variables, and lifestyle factors in shaping individual health outcomes. These findings contribute to our understanding of the complex interplay between various factors and AL, paving the way for future research aimed at developing targeted interventions and public health strategies to improve health and well-being.

## Supporting information

**S1 Table. Descriptive summary for the whole sample.**
(DOCX)

**S2 Table. Counts of missing data points for biomarkers at baseline and follow-up examinations.**
(DOCX)

**S3 Table. Comparison of different calculation methods for AL scores Mean [95% CI].**
(DOCX)

**S4 Table characteristics of time-variant variables at first follow-up (sub-cohort).**
(DOCX)

**S5 Table. AL scores stratified by age cohorts for men.**
(DOCX)

**S6 Table. AL scores stratified by age cohorts for women.**
(DOCX)

## Acknowledgments

We are grateful to all of the participants in the CARLA-cohort, to the survey staff and research nurses who made this study possible. Further, we want to thank Alexander Kluttig and Lena Minning for the additional work in data preparation.

## Author contributions

**Conceptualization:** Eric Priedemann, Rafael Mikolajczyk, Amand Führer.

**Data curation:** Eric Priedemann, Alexander Kluttig.

**Formal analysis:** Eric Priedemann.

**Methodology:** Eric Priedemann, Rafael Mikolajczyk, Amand Führer.

**Project administration:** Alexander Kluttig.

**Resources:** Frank Bernhard Kraus.

**Supervision:** Rafael Mikolajczyk, Amand Führer.

**Validation:** Frank Bernhard Kraus, Amand Führer.

**Visualization:** Eric Priedemann.

**Writing – original draft:** Eric Priedemann.

**Writing – review & editing:** Eric Priedemann, Alexander Kluttig, Frank Bernhard Kraus, Daniel Sedding, Rafael Mikolajczyk, Amand Führer.

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
