## [Decision Letter · Decision Letter 0]

30 Dec 2024

PONE-D-24-31153Allostatic Load and its determinants in a German sample – Results from the Carla cohort.PLOS ONE

Dear Dr. Priedemann,

Thank you for submitting your manuscript to PLOS ONE. After careful consideration, we feel that it has merit but does not fully meet PLOS ONE’s publication criteria as it currently stands. Therefore, we invite you to submit a revised version of the manuscript that addresses the points raised during the review process. Please submit your revised manuscript by Feb 13 2025 11:59PM. If you will need more time than this to complete your revisions, please reply to this message or contact the journal office at plosone@plos.org . Please include the following items when submitting your revised manuscript:

We look forward to receiving your revised manuscript.

Kind regards,

Andrey I Egorov

Academic Editor

PLOS ONE

Journal Requirements: When submitting your revision, we need you to address these additional requirements. 1. Please ensure that your manuscript meets PLOS ONE's style requirements, including those for file naming. The PLOS ONE style templates can be found at https://journals.plos.org/plosone/s/file?id=wjVg/PLOSOne_formatting_sample_main_body.pdf and https://journals.plos.org/plosone/s/file?id=ba62/PLOSOne_formatting_sample_title_authors_affiliations.pdf 2. We note that the grant information you provided in the ‘Funding Information’ and ‘Financial Disclosure’ sections do not match.  When you resubmit, please ensure that you provide the correct grant numbers for the awards you received for your study in the ‘Funding Information’ section. 3. Thank you for stating the following financial disclosure: "We acknowledge the financial support of the Open Access Publication Fund of the Martin-Luther 498 University Halle-Wittenberg." Please state what role the funders took in the study.  If the funders had no role, please state: ""The funders had no role in study design, data collection and analysis, decision to publish, or preparation of the manuscript."" If this statement is not correct you must amend it as needed. Please include this amended Role of Funder statement in your cover letter; we will change the online submission form on your behalf. 4. PLOS requires an ORCID iD for the corresponding author in Editorial Manager on papers submitted after December 6th, 2016. Please ensure that you have an ORCID iD and that it is validated in Editorial Manager. To do this, go to ‘Update my Information’ (in the upper left-hand corner of the main menu), and click on the Fetch/Validate link next to the ORCID field. This will take you to the ORCID site and allow you to create a new iD or authenticate a pre-existing iD in Editorial Manager. 5. Your ethics statement should only appear in the Methods section of your manuscript. If your ethics statement is written in any section besides the Methods, please move it to the Methods section and delete it from any other section. Please ensure that your ethics statement is included in your manuscript, as the ethics statement entered into the online submission form will not be published alongside your manuscript. 6. We notice that your supplementary tables are included in the manuscript file. Please remove them and upload them with the file type 'Supporting Information'. Please ensure that each Supporting Information file has a legend listed in the manuscript after the references list.

Reviewers' comments:

Reviewer's Responses to Questions

**Comments to the Author**

1. Is the manuscript technically sound, and do the data support the conclusions?

Reviewer #1: Partly

Reviewer #2: Yes

2. Has the statistical analysis been performed appropriately and rigorously? 

Reviewer #1: I Don't Know

Reviewer #2: Yes

3. Have the authors made all data underlying the findings in their manuscript fully available?

Reviewer #1: Yes

Reviewer #2: Yes

4. Is the manuscript presented in an intelligible fashion and written in standard English?

Reviewer #1: Yes

Reviewer #2: Yes

5. Review Comments to the Author

Reviewer #1: The article "Allostatic Load and its Determinants in a German Sample - Results from the Carla Cohort" aims to describe changes in allostatic load over time and to identify longitudinal determinants of allostatic load. The allostatic load index includes information on metabolic, immune, cardiovascular and anthropometric measures. The results show longitudinal changes in allostatic load among men. Allostatic load was associated with occupational status and physical activity. The article includes many analyses, but should be revised to focus more on the research objective.

Lines 42-44: The conclusion does not follow the results - what public health interventions are meant?

The background section gives a good overview of the existing literature, but needs to be rewritten to focus on the research aim.

Is the first aim the description of changes over time or the trend of AL over time in different age groups?

Lines 160-169: Please include a flow chart

Table 1: Add "Biomarkers used to calculate AL score for comparison over time" to clarify that cut points are only used in this version of the score.

Statistical Analysis: Please add information about the statistical analysis software used

Lines 261-265: Add more information about the type of statistical model used. Is the baseline level of the AL score included as an independent variable? Explain why you used different models rather than one model that included change in score and determinants over time? Especially since lifestyle factors and medication use changed over the observation period (lines 283-297).

Table 3: Are the categories shown baseline covariates?

Lines 363-364: this was not a result of this study and very speculative. Could be a selection bias due to loss to follow-up.

Strenghts and Limitations: Are the AL scores based on single measures at 1 time point? Then add information on intra-individual variability of serum lipids and blood pressure and thus misclassification bias.

Reviewer #2: Authors assessed the allostatic load from the German Carla cohort.

This study was properly conducted and evaluated.

However, biomarkers used in the study was too basic, hence if plasma samples are available, authors should perform latest biomarkers such as following cortisol, VILIP-1 and MBP.

6. PLOS authors have the option to publish the peer review history of their article (what does this mean? ). If published, this will include your full peer review and any attached files.

**Do you want your identity to be public for this peer review?** For information about this choice, including consent withdrawal, please see our Privacy Policy .

Reviewer #1: **Yes: ** Julia Truthmann

Reviewer #2: No

---

## [Author Response · Author response to Decision Letter 1]

5 Feb 2025

We sincerely thank you for your thorough review and valuable feedback on our manuscript. Your insightful comments and suggestions have significantly helped us to refine and improve the quality of our work. We deeply appreciate the time and effort you invested in reviewing our paper, and we are grateful for your constructive input. Below, we address each of your comments in detail.

Sincerely,

Eric Priedemann for the authors

Reviewer #1

Comment 1: Lines 42-44: The conclusion does not follow the results - what public health interventions are meant? - Our current findings do not allow for specific public health interventions but indicate subgroup-specific influences of chronic stress on aging processes. As you correctly pointed out, the previously used phrasing was overly broad and has been revised accordingly.

Comment 2: The background section gives a good overview of the existing literature, but needs to be rewritten to focus on the research aim. - Your comment helped us improve the transition from the current state of research on allostatic load to our specific research question. This transition has been supplemented with paragraphs highlighting the research gaps in the existing literature and leading into our research questions. We hope this serves as a helpful link that makes it easier for readers to understand the outlined issue.

Comment 3: Is the first aim the description of changes over time or the trend of AL over time in different age groups? – To clarify our objective, the first research question has been split into two questions:

1. Do mean AL scores change in the CARLA-cohort over the study period for CARLA?

We aim to investigate how the AL scores change within the cohort over the study period.

2. Do mean AL scores change differently for men and women and within age-groups?

We also aim to examine whether the observed changes in AL scores at the study time point differ between sex and age cohorts (measured at baseline). Specifically, we are investigating whether participants under 55 years of age at baseline show different changes in AL scores compared to participants aged 65-70 years at baseline.

Comment 4: Lines 160-169: Please include a flow chart - A flow chart has been added (Fig. 1).

Comment 5: Table 1: Add "Biomarkers used to calculate AL score for comparison over time" to clarify that cut points are only used in this version of the score. – We included this specification in the heading of Table 1.

Comment 6: Statistical Analysis: Please add information about the statistical analysis software used – We used the RStudio software environment for statistical analyses. This information was added at the end of the method section.

Comment 7: Lines 261-265: Add more information about the type of statistical model used. Is the baseline level of the AL score included as an independent variable? Explain why you used different models rather than one model that included change in score and determinants over time? Especially since lifestyle factors and medication use changed over the observation period (lines 283-297). –

As this paper specifically aims to explore the association between the independent variables and AL scores within our cohort at different follow-ups, we rely on the baseline values of the predictors.

We have explicitly decided against adjusting for the baseline level of AL based on the following four reasons:

1. Adjusting for the baseline value of a dependent variable carries the risk of collider bias. In this case, one controls for a variable that may be influenced both by the independent variables and by unobserved factors (1). In the present model, this could lead to an overestimation of the associations between the independent variables and AL.

2. As we are analyzing a cohort with a high average age, where some independent variables (e.g., educational attainment) exerted their effects long before the baseline examination, the AL at baseline partially reflects the associations with these independent variables (2). Adjusting for AL at baseline would thus underestimate these associations.

3. Adjusting for the AL at baseline would measure only the associations between the independent variables and AL that go beyond the effect of AL at baseline (3). This would capture changes in AL, which is not the research focus of the present study.

4. A correlation between the baseline value of a dependent variable and subsequent measurements is possible and could lead to overfitting, thereby impairing the predictive power and robustness of the model (4).

Furthermore, your comment points to another very interesting question regarding the trajectories of AL and potential factors influencing the baseline level as well as the increase or decrease of AL. Due to the significant scope of these different questions, we had to refrain from addressing them within the scope of this paper. A second paper, which addresses this intriguing question, is currently under review (5).

Comment 8: Table 3: Are the categories shown baseline covariates? - In Table 3, we used baseline values of the covariates. As such, it represents a stratification of AL scores based on the covariates measured at baseline. We have added this information in the table legend for clarification.

Comment 9: Lines 363-364: this was not a result of this study and very speculative. Could be a selection bias due to loss to follow-up. – In the revised version, we acknowledge the lifestyle changes that occurred during the study period, which could have potentially led to an improvement in the AL scores. However, we emphasize that, due to the low retention rate, no conclusions can be made regarding any association.

Comment 10: Strenghts and Limitations: Are the AL scores based on single measures at 1 time point? Then add information on intra-individual variability of serum lipids and blood pressure and thus misclassification bias. - The venous blood samples were taken from non-fasting participants once at each examination. In the revised version, we point out the resulting intraindividual variations; however, due to the completion of the studies, we are unable to make any methodological changes. In addition to this natural fluctuation, which can lead to misclassification of participants (as you rightly pointed out), significant intraindividual variation is also a risk factor for cardiovascular endpoints, meaning a "falsely high" value also represents a part of the cardiovascular risk and chronic stress of the body. Moreover, the use of lipid-lowering medications has different effects on these fluctuations, as it is known, for example, that the use of long-acting statins or PCSK-9 inhibitors results in less fluctuation of serum-lipids than shorter-acting statins (6). Since this precise specification is not evident from the study data, and the value of these different influences on AL cannot be reliably quantified, we believe that pointing out the existing fluctuations is sufficient. We are very grateful for this suggestion and appreciate this detailed scientific discourse, which contributes to a better understanding of AL.

Reviewer #2

Comment 11: Authors assessed the allostatic load from the German Carla cohort.

This study was properly conducted and evaluated.

However, biomarkers used in the study was too basic, hence if plasma samples are available, authors should perform latest biomarkers such as following cortisol, VILIP-1 and MBP. – Thank you for your helpful suggestion regarding our work. In selecting biomarkers for AL, we focused on including markers from different systems. However, we were constrained by the availability of data and the need for consistent and comprehensive collection of these biomarkers across all the follow-ups used. The score we employed reflects the maximum overlap given these limitations and cannot be expanded further due to the unavailability of additional biomarkers.

Literature used for the response to the reviewers

1. Hernan MA, Monge S. Selection bias due to conditioning on a collider. BMJ. 2023;381:1135.

2. Glymour MM, Weuve J, Berkman LF, Kawachi I, Robins JM. When is baseline adjustment useful in analyses of change? An example with education and cognitive change. Am J Epidemiol. 2005;162(3):267-78.

3. (CHMP) CfMPfHU. Guideline on adjustment for baseline covariates in clinical

trials. European Medicines Agency Science Medicines Health; London2015.

4. Babyak MA. What you see may not be what you get: a brief, nontechnical introduction to overfitting in regression-type models. Psychosom Med. 2004;66(3):411-21.

5. Priedemann EN, Kluttig A, Kraus FB, Sedding D, Mikolajczyk R, Führer A. Association between Social Mobility, Social Status and trajectories of Allostatic Load – A longitudinal analysis of the Carla cohort. Stress and Health. in submission.

6. Simpson WG. Biomarker variability and cardiovascular disease residual risk. Curr Opin Cardiol. 2019;34(4):413-7.

---

## [Decision Letter · Decision Letter 1]

4 Mar 2025

Allostatic Load and its determinants in a German sample – Results from the Carla cohort.

PONE-D-24-31153R1

Dear Dr. Priedemann,

We’re pleased to inform you that your manuscript has been judged scientifically suitable for publication and will be formally accepted for publication once it meets all outstanding technical requirements.

Kind regards,

Andrey I Egorov

Academic Editor

PLOS ONE

Reviewers' comments:

Reviewer's Responses to Questions

**Comments to the Author**

1. If the authors have adequately addressed your comments raised in a previous round of review and you feel that this manuscript is now acceptable for publication, you may indicate that here to bypass the “Comments to the Author” section, enter your conflict of interest statement in the “Confidential to Editor” section, and submit your "Accept" recommendation.

Reviewer #1: All comments have been addressed

2. Is the manuscript technically sound, and do the data support the conclusions?

Reviewer #1: Yes

3. Has the statistical analysis been performed appropriately and rigorously? 

Reviewer #1: Yes

4. Have the authors made all data underlying the findings in their manuscript fully available?

Reviewer #1: Yes

5. Is the manuscript presented in an intelligible fashion and written in standard English?

Reviewer #1: Yes

6. Review Comments to the Author

Reviewer #1: (No Response)

7. PLOS authors have the option to publish the peer review history of their article (what does this mean? ). If published, this will include your full peer review and any attached files.

**Do you want your identity to be public for this peer review?** For information about this choice, including consent withdrawal, please see our Privacy Policy .

Reviewer #1: **Yes: ** Julia Truthmann

---

## [Editor Report · Acceptance letter]

PONE-D-24-31153R1

PLOS ONE

Dear Dr. Priedemann,

I'm pleased to inform you that your manuscript has been deemed suitable for publication in PLOS ONE. Congratulations! Your manuscript is now being handed over to our production team.

Kind regards,

on behalf of

Dr. Andrey I Egorov

Academic Editor

PLOS ONE